# A Mean-Field Game Control for Large-Scale Swarm Formation Flight in Dense Environments

**DOI:** 10.3390/s22145437

**Published:** 2022-07-21

**Authors:** Guofang Wang, Wang Yao, Xiao Zhang, Ziming Li

**Affiliations:** 1School of Mathematical Sciences, Beihang University, Beijing 100191, China; wangguofang@buaa.edu.cn (G.W.); xiao.zh@buaa.edu.cn (X.Z.); zimingli@buaa.edu.cn (Z.L.); 2Key Laboratory of Mathematics, Informatics and Behavioral Semantics, Ministry of Education, Beijing Advanced Innovation Center for Big Data and Brain Computing, Beihang University, Beijing 100191, China; 3Peng Cheng Laboratory, Shenzhen 518055, China; 4Institute of Artificial Intelligence, Beihang University, Beijing 100191, China

**Keywords:** large-scale UAV swarm, multiagent coordination, formation flight, collision avoidance, MFG control, distributed decisions

## Abstract

As an important part of cyberphysical systems (CPSs), multiple aerial drone systems are widely used in various scenarios, and research scenarios are becoming increasingly complex. However, planning strategies for the formation flying of aerial swarms in dense environments typically lack the capability of large-scale breakthrough because the amount of communication and computation required for swarm control grows exponentially with scale. To address this deficiency, we present a mean-field game (MFG) control-based method that ensures collision-free trajectory generation for the formation flight of a large-scale swarm. In this paper, two types of differentiable mean-field terms are proposed to quantify the overall similarity distance between large-scale 3-D formations and the potential energy value of dense 3-D obstacles, respectively. We then formulate these two terms into a mean-field game control framework, which minimizes energy cost, formation similarity error, and collision penalty under the dynamical constraints, so as to achieve spatiotemporal planning for the desired trajectory. The classical task of a distributed large-scale aerial swarm system is simulated by numerical examples, and the feasibility and effectiveness of our method are verified and analyzed. The comparison with baseline methods shows the advanced nature of our method.

## 1. Introduction

As an important part of cyberphysical systems (CPSs), multiple aerial drone systems have shown great advantages and value in performing tasks such as environmental reconnaissance, information transmission & reception, and target tracking in the fields of disaster rescue, geological exploration, and smart cities [1]. Compared with common multi unmanned aerial vehicle (UAV) systems, hundreds or thousands of larger-scale UAV swarms have better maneuverability performance and mission realization efficiency [2]. When the drone swarm performs actual missions, it will fly in formation as needed. In the process of approaching the target, it not only needs to avoid obstacles in time but also needs to consider the trajectory collision avoidance between the drones in the swarm. As a result, the drone swarm requires a large amount of communication and computing resources to guarantee real-time communication information interaction and perception information calculation, which will grow exponentially with the scale.

Research work on the flight control of aerial swarms is extensive. A large number of research studies focus on the formation flight, trajectory planning, and obstacle avoidance of the swarm, and their combination forms are more common. Specifically, Zhang et al. [3] proposed a formation control method based on the communication consensus mechanism and leader–follower strategy, which realized the efficient flight of multiple UAVs in formation around a single static obstacle. In the same year, Zhang et al. [4] proposed an obstacle avoidance control algorithm based on virtual structure and 3-D spatial leader–follower strategy, which realized a triangular formation of multiple UAVs flying through multiple static obstacles with high efficiency. Shao et al. [5] proposed an improved particle swarm optimization algorithm for formation-obstacle avoidance-trajectory planning in a 3-D environmen and completed an experiment in which multiple UAVs formed a formation after avoiding the static cone and ellipsoid obstacles. Combined with an improved artificial potential field method (APF), Wu et al. [6] performed a swarm control for multiple UAVs based on a virtual core structure, solved the local minima problem in APF, and successfully achieved the swarm control of multiple UAVs and the flexible obstacle avoidance of adaptive formation flight in an urban environment. In 2021, the Fei Gao team [7] proposed a distributed swarm trajectory optimization method for formation flight in a dense environment, which realizes multiple UAVs maintain formation flight through unknown obstacle-rich scenarios. Peng et al. [8] put forward a perception sharing and swarm trajectory global optimal algorithm, which experimentally showed that the success rate of obstacle avoidance in a dense obstacle environment could reach at least 80%. In the latest research so far, the formation-obstacle avoidance scenarios of multi-UAV systems have become more and more complex, and the continuously developed optimal control algorithms provide decision-making guarantees for mission flight in the corresponding scenarios. However, the current situation shows that the research schemes in the above literature are all limited by the large-scale of UAVs because the traditional optimization and control technologies deal with the dynamic interaction between individuals, respectively. With the increase of the number of UAVs, the difficulty of solving the optimal strategy will increase significantly.

For the large-scale UAVs flight control problem, some researchers have contributed interesting work. For example, [9] used the mean-field game (MFG) model to solve the real-time flight control problem of large-scale UAVs on the 2-D plane, and the proposed algorithm effectively realizes interaircraft collision avoidance and reduces the energy consumption of UAVs. Shiri et al. [10] combined a federal learning method that can share the parameters of neural network models on UAVs and completed a simulation experiment based on the MFG model for large-scale-UAV swarms flying to the destination under 2-D conditions. Chen et al. [11] proposed a mean-field trust-region policy optimization control method based on multi-agent reinforcement learning, which solved the problem of limited communication range for large-scale UAV flight control. Xu et al. [12] proposed a dual-fields approach, in which the improved APF method overcomes the failure of traditional APF in collision avoidance; the MFG method greatly reduces the communication interference of the UAV group, couples these two methods to adjust the flight trajectory and power consumption, and realizes the flight obstacle avoidance of large-scale UAVs in multiple static obstacles with height differences. Gao et al. [13] proposed an energy-efficient velocity control algorithm for a large number of UAVs based on MFG, which expressed the speed control of large-scale UAVs as a differential game and used the original double mixed gradient method to solve the problem. Their experiments completed that large-scale UAVs bypassed static obstacles in the case of 2-D while minimizing energy consumption. The above documents all use the mean-field method to convert the traditional control 1-vs.-N game into a 1-vs.-1 game, which greatly reduces the interaction frequency of agent systems. Thus, they can deal with large-scale mission scenarios to effectively reduce the communication resources and flight energy consumption. However, it should be pointed out that the state quantities in their numerical solution of the mean-field are all low-dimensional, and the relevant numerical solution problems can be solved using the grid-based method. In terms of solving high-dimensional MFGs, [14] perfectly avoids spatial grids of high-dimensional MFGs based on the machine learning framework, but their work is limited to the deterministic setting (σ = 0). Ref. [15] is the first document to solve high-dimensional MFGs under a random setting (σ > 0), which exploits the natural connection between MFGs and generative adversarial neural networks (GANs) [16]. Ref. [17] further proposed a numerical solution for solving multi-population high-dimensional stochastic MFGs, a coupled alternating neural network (CA-Net). The numerical experiments completed the coordinated flight of large-scale multipopulation quadcopters to the destination and realize the intergroup interaction and intragroup collision avoidance. The MF method, MFG model, and its numerical solution methods gathered from these documents provide a preliminary solution idea for the flight control problem of large-scale aerial drones in 3-D scenes. However, the current situation shows that these large-scale control research schemes are limited by complex scenarios because most of the state quantities in MF terms are directly used for real-time error measurement. With the complexity of application scenarios, a large number of UAV states may change frequently, resulting in a significant decline in mission success rate.

To summarize, a large-scale agent trajectory planning method that can effectively manage both formation and obstacle avoidance in dense environments is lacking in the literature. On the one hand, obstacle-dense avoidance control for aerial swarm navigation in formation needs to break through the large-scale restrictions; on the other hand, large-scale UAV flight control methods cannot be directly applied to complex scenarios. In practice, a single agent will stay away from obstacles for safety. However, when formation imposes agents tracking targets, it may oppose obstacle avoidance, that is, sometimes formation and obstacle avoidance are contradictory; large-scale itself has the difficulty of communication and calculation, which is exacerbated by frequent formation–neighbor communication and obstacle avoidance environment interactions. How to systematically trade off the conflicting requirements of large-scale, formation, and obstacle avoidance is the key point to accomplishing large-scale noncolliding formation flights.

To bridge the gap, we propose a large-scale agent trajectory optimization method capable of navigating large-scale swarms in formations while avoiding dense obstacles. First, extending the formation method in [7], we model the formation using undirected graphs based on probability densities and define a mean-field-based differentiable metric that assesses the difference between large-scale formation shapes in 3-D workspaces. Our formation similarity metric also retains the translation, rotation, and scale invariance of the formation methods in [7], enabling quantitative evaluation of the overall performance of large-scale formation maintenance. To solve the communication and computing difficulties caused by complex scenarios of large-scale multiagent systems and formulate the trade-offs between formation and obstacle avoidance in large-scale contexts, we design a mean-field game control framework that simultaneously satisfies the dynamic constraints and minimizes energy cost, formation similarity error, and collision penalty. The latter two of them serve as mean-field terms directly related to complex environments, and the functional form we define provides greater flexibility for agent maneuvering. Finally, to verify the effectiveness and practicability of the method, we conduct extensive simulation experiments of distributed aerial large-scale UAVs.

Our main contributions are summarized as:A differentiable graph-theory-based mean-field term that quantifies the similarity distance between large-scale three-dimensional formations; a differentiable ellipsoid-based mean-field term that inscribes the potential energy value of dense three-dimensional obstacles.A general control framework for complex scenarios of large-scale multiagent systems—mean-field game control, which jointly takes the amount of communication and computation, operating energy consumption, formation similarity, and obstacle avoidance into account.A series of simulations with a distributed large-scale aerial swarm system validates the efficiency and robustness of our method. The comparison with baseline methods shows the advanced nature of our method.

## 2. Two Types of Mean-Field Terms

An aerial swarm formed by a large number of UAVs is considered in this paper, which is traversing an obstacle-rich area in an expected formation. Herein, two types of differentiable mean-field terms for formation similarity metric and obstacle potential energy value engraving are constructed in the following two subsections.

### 2.1. Formation Mean-Field Term

The large-scale system N with *N* agents have a probability distribution of the spatial state x varying with time *t*, that is, the mean field ρ(x,t), where $x=\left[x_{1},x_{2},x_{3},...,x_{n}\right]\in R^{n}∼\rho$. Usually the position vector $p=\left[x,y,z\right]=\left[x_{1},x_{2},x_{3}\right]$. A formation of *N* agents is modeled by an undirected graph G=(V,E), where V:={1,2,…,N} is the set of vertices, and E⊂V×V is the set of edges [7]. In graph G, the vertex *i* represents the $i^{\mathrm{th}\;}$ agent with position vector $p_{i}=\left[x_{i},y_{i},z_{i}\right]\in R^{3}$. An edge $e_{ij}\in E$ that connects vertex i∈V and vertex j∈V means the agent *i* and *j* can measure the geometric distance between each other. In this work, each agent communicates with all other agents, thus, the formation graph G is complete. Each edge of the graph G is associated with a non-negative number as a weight. The weight of edge $e_{ij}$ is given by
(1)$$w_{ij}={∥p_{i}-p_{j}∥}^{2},\;\left(i,j\right)\in E,$$
where ∥·∥ denotes the Euclidean norm. Now, the adjacency matrix $A=\left(A_{ij}\right)\in R^{N\times N}$ and degree matrix $D=\left(D_{ij}\right)\in R^{N\times N}$ of the formation G are determined,
(2)$$A_{ij}=w_{ij},$$
(3)$$D_{ij}=\left\{\begin{array}{cc} \sum_{j}A_{ij} & \;\mathrm{if}\;i=j\\ 0 & \;\mathrm{otherwise}\;. \end{array}\right.$$

Thus, the corresponding Laplacian matrix is given by
(4)L=D−A.

With the above matrices, the symmetric normalized Laplacian matrix of the graph G is defined as
(5)$$\hat{L}=D^{-1/2}{\mathrm{LD}}^{-1/2}=I-D^{-1/2}{\mathrm{AD}}^{-1/2},$$
where $I\in R^{N\times N}$ is the identity matrix.

As a graph representation matrix, Laplacian contains information about the graph structure [18]. To achieve the expected formation for the large-scale swarm, we propose a formation similarity metric as
(6)$$F^{1}\left(\rho \left(x,t\right)\right)=\int_{\Omega }\int_{\Omega }f^{1}\left(x,\hat{x}\right)d\rho \left(x,t\right)d\rho \left(\hat{x},t\right),$$
(7)$$f^{1}\left(x,\hat{x}\right)={∥\hat{L}\left(x\right)-{\hat{L}}_{e}\left(\hat{x}\right)∥}_{F}^{2}=\mathrm{tr}\left\{{\left(\hat{L}-{\hat{L}}_{e}\right)}^{T}\left(\hat{L}-{\hat{L}}_{e}\right)\right\},$$
where Ω denotes state space of x, tr{·} is the trace of a matrix, $\hat{L}$ is the symmetric normalized Laplacian of the current swarm formation, and ${\hat{L}}_{e}$ is the counterpart of the expected formation. Frobenius norm ${∥\cdot ∥}_{F}$ is used in our distance metric.

**Remark** **1.**
*The function $F^{1}$ is invariant to the translation, rotation, and scale of the formation because $F^{1}$ is an integral of $f^{1}$. $f^{1}$ is natively invariant to translation and rotation of the formation since the corresponding graph is weighted by the absolute distance between agent positions; scaling invariance is achieved by normalizing graph Laplacian with the degree matrix in (Equation 5) [7].*


### 2.2. Obstacle Mean-Field Term

The large-scale system N with *N* UAVs needs to traverse an obstacle-rich area. There are *K* static obstacles. The position of UAV is available. We assume that obstacles can be detected by the UAV vision sensor. In a 3-D urban environment, without loss of generality, obstacles can be considered rectangular solids $\Omega_{obs,k}=\{\left(x,y,z\right)||x-x_{0,k}|\leq v_{1,k},|y-y_{0,k}|\leq v_{2,k},|z-z_{0,k}\left|\leq v_{3,k}\right\}$, k=1,...,K where $\left(x_{0,k},y_{0,k},z_{0,k}\right)$ is the center of corresponding obstacle, $\left(\pm v_{1,k},\pm v_{2,k},\pm v_{3,k}\right)$ are its vertices that are parallel to the x,y,z-axis, respectively. For training, we formulate obstacles as
(8)$$\Omega_{obs,trn,k}:\;Q_{k}\left(x,y,z\right)=\frac{1}{3v_{1,k}^{2}}{\left(x-x_{0,k}\right)}^{2}+\frac{1}{3v_{2,k}^{2}}{\left(y-y_{0,k}\right)}^{2}+\frac{1}{3v_{3,k}^{2}}{\left(z-z_{0,k}\right)}^{2}\leq 1.1\;k=1,...,K.$$

We encode ellipsoidal repulsion of obstacles for the generic UAV as
(9)$$f^{2}\left(x\right)=\left\{\begin{array}{cc} \sum_{k=1}^{K}\gamma_{obs,k}\frac{1}{Q_{k}} & \;\mathrm{if}\;x\in \Omega_{obs,trn}\\ 0 & \;\mathrm{otherwise}\;, \end{array}\right.$$
where $\gamma_{obs,k}$, k=1, ⋯, K, is repulsive force gain coefficient between the generic UAV and other obstacles, $\Omega_{obs,trn}={⋃}_{k=1}^{K}\Omega_{obs,trn,k}$.

**Remark** **2.**
*The cuboid obstacle repulsion is encoded as the unit circumscribed ellipsoid repulsion making this function differentiable [19]. $\Omega_{obs,trn}$ contains an obstacle radial bound ten percent more than in circumscribed ellipsoid of $\Omega_{obs}$ because we found this additional training buffer alleviates collisions during validation. By training with ellipsoidal repulsion, which has gradient information within the obstacles, we incentivize the model to learn trajectories avoiding the obstacles [20].*


To achieve all obstacles avoidance for the large-scale swarm, we propose an obstacle potential energy function as
(10)$$F^{2}\left(\rho \left(x,t\right)\right)=\int_{\Omega }f^{2}\left(x\right)\rho \left(x,t\right)dx.$$

**Remark** **3.**
*The function $F^{2}$ can avoid dense obstacles for large-scale swarms, which benefits from the proper construction of 3-D obstacles on the one hand, and the organic combination of obstacle potential and mean-field method on the other hand.*


## 3. A Mean-Field Game Control Framework for Complex Scenarios

In this section, we find the optimal state control for large-scale UAVs which can systematically weigh the energy consumption, formation similarity, and obstacle avoidance of each UAV when it travels complex scenarios. Specifically, we first formulate the optimal control problem for a single UAV. The interaction relationship between drones forms an N-agent noncooperative differential game. As the number of UAVs grows large, the complexity of solving the differential game in the optimal control problem can increase significantly as traditional methods need to separately deal with the increasing interactions. Therefore, we reformulate the large-scale UAVs optimal state control problem in complex scenarios as a mean-field game control problem. Subsequently, we propose a generative-adversarial-network (GAN)-based algorithm to solve the mean-field game.

### 3.1. Single-UAV Optimal Control Problem

We consider that the continuous-time state x of the *i*th UAV has the dynamic–kinematic equation of the following form
(11)$$dx_{i}=h_{i}\left(x_{i},x_{i}^{-},u_{i}\right)dt+\sigma d\omega_{i}\left(t\right),$$
where $x_{i}^{-}$ denotes the states of all other UAVs, $u_{i}\in U_{i}$ is the control input (strategy). In the stochastic case, we add a noise term to the dynamics, σ is a volatility term—a fixed coefficient matrix, and $\omega_{i}\left(t\right)$ denotes a Wiener process (standard Brownian motion), which is identical and independent among all UAVs. The interpretation here is that we are modeling the situation when the quadcopter suffers from noisy measurements [15]. For example, under wind perturbations, σ can be the covariance matrix of the wind velocity [10,13].

Without loss of generality, denoting gravity as *g*, the acceleration of a UAV with mass *m* can be written as
(12)$$\left\{\begin{array}{c} \ddot{x_{i}}=\frac{u_{i}}{m}\left(\sin\left(\psi_{i}\right)\sin\left(\varphi_{i}\right)+\cos\left(\psi_{i}\right)\sin\left(\theta_{i}\right)\cos\left(\varphi_{i}\right)\right)\\ \ddot{y_{i}}=\frac{u_{i}}{m}\left(-\cos\left(\psi_{i}\right)\sin\left(\varphi_{i}\right)+\sin\left(\psi_{i}\right)\sin\left(\theta_{i}\right)\cos\left(\varphi_{i}\right)\right)\\ \ddot{z_{i}}=\frac{u_{i}}{m}\cos\left(\theta_{i}\right)\cos\left(\varphi_{i}\right)-g\\ \ddot{\psi_{i}}={\tau_{\psi }}_{i}\\ \ddot{\theta_{i}}={\tau_{\theta }}_{i}\\ \ddot{\varphi_{i}}={\tau_{\varphi }}_{i} \end{array}\right.,$$
where (x,y,z) is the spatial position of the UAV, (ψ,θ,φ) is the angular orientation with corresponding torques $\tau_{\psi },\tau_{\theta },\tau_{\varphi }$, and *u* is the main thrust directed out of the bottom of the aircraft [21]. To fit a control framework, the above second-order system can be turned into a first-order system
(13)$$\dot{x_{i}}=h_{i}\left(x_{i},x_{i}^{-},u_{i}\right)⟹\left\{\begin{array}{c} \dot{x_{i}}={v_{x}}_{i}\\ \dot{y_{i}}={v_{y}}_{i}\\ \dot{z_{i}}={v_{z}}_{i}\\ \dot{\psi_{i}}={v_{\psi }}_{i}\\ \dot{\theta_{i}}={v_{\theta }}_{i}\\ \dot{\varphi_{i}}={v_{\varphi }}_{i}\\ {\dot{v_{x}}}_{i}=\frac{u_{i}}{m}\left(\sin\left(\psi_{i}\right)\sin\left(\varphi_{i}\right)+\cos\left(\psi_{i}\right)\sin\left(\theta_{i}\right)\cos\left(\varphi_{i}\right)\right)\\ {\dot{v_{y}}}_{i}=\frac{u_{i}}{m}\left(-\cos\left(\psi_{i}\right)\sin\left(\varphi_{i}\right)+\sin\left(\psi_{i}\right)\sin\left(\theta_{i}\right)\cos\left(\varphi_{i}\right)\right)\\ {\dot{v_{z}}}_{i}=\frac{u_{i}}{m}\cos\left(\theta_{i}\right)\cos\left(\varphi_{i}\right)-g\\ {\dot{v_{\psi }}}_{i}={\tau_{\psi }}_{i}\\ {\dot{v_{\theta }}}_{i}={\tau_{\theta }}_{i}\\ {\dot{v_{\varphi }}}_{i}={\tau_{\varphi }}_{i} \end{array}\right.,$$
where $x={\left[x,y,z,\psi ,\theta ,\varphi ,v_{x},v_{y},v_{z},v_{\psi },v_{\theta },v_{\varphi }\right]}^{T}\in R^{12}$ is the state with velocities v, and $u={\left[u,\tau_{\psi },\tau_{\theta },\tau_{\varphi }\right]}^{T}\in R^{4}$ is the control.

For the *i*th UAV, we consider the cost function of the following form
(14)$$J_{i}\left(x_{i},x_{i}^{-},u_{i}\right)=\int_{0}^{T}L_{i}\left(x_{i},u_{i}\right)+F_{i}\left(x_{i},x_{i}^{-}\right)dt+G_{i}\left(x_{i}^{T}\right),$$
where $L_{i}$ is a running cost incurred by the *i*th UAV based solely on its actions, $F_{i}$ is a running cost incurred by the *i*th UAV based on its interactions with the rest of the swarm, and $G_{i}$ is a terminal cost incurred by the *i*th UAV based on its final state.

The running cost $L_{i}$ is given by
(15)$$L_{i}\left(x_{i},u_{i}\right)=c_{1}{∥u_{i}\left(x_{i},t\right)∥}^{2},$$
where $c_{1}$ is a constant. The running cost $L_{i}$ denotes the control effort implemented by the *i*th UAV. The terminal cost $G_{i}$ is given by
(16)$$G_{i}\left(x_{i}^{T}\right)=c_{2}{∥x_{i}^{T}-x_{0}∥}_{2},$$
where $c_{2}$ is a constant, $x_{i}^{T}$ is the final state of the *i*th UAV, and $x_{0}$ is the target state on which we want the UAVs to reach the objective. The terminal cost $G_{i}$ denotes the distance between UAV *i*’s target state and the desired state. The interaction cost $F_{i}$ is given by
(17)$$F_{i}\left(x_{i},x_{i}^{-}\right)=l_{1}F_{i}^{1}\left(x_{i},x_{i}^{-},t\right)+l_{2}F_{i}^{2}\left(x_{i},x_{i}^{-},t\right)+l_{3}F_{i}^{3}\left(x_{i},x_{i}^{-},t\right),$$
where $l_{1}$, $l_{2}$, $l_{3}$ are constants, $F_{i}^{1}\left(x_{i},x_{i}^{-},t\right)$ is UAV *i*’s formation cost,
(18)$$F_{i}^{1}\left(x_{i},x_{i}^{-},t\right)=\frac{1}{N}\sum_{i=1}^{N}\frac{1}{N}\sum_{j=1}^{N}{∥\hat{L}\left(x_{i},x_{i}^{-},t\right)-{\hat{L}}_{e}\left({\hat{x}}_{j},{\hat{x}}_{j}^{-},t\right)∥}_{F}^{2},$$

$F_{i}^{2}\left(x_{i},x_{i}^{-},t\right)$ is UAV *i*’s obstacle avoidance cost,
(19)$$F_{i}^{2}\left(x_{i},x_{i}^{-},t\right)=\frac{1}{N}\sum_{i=1}^{N}\left\{\begin{array}{cc} \sum_{k=1}^{K}\gamma_{obs,k}\frac{1}{Q_{k}\left(x_{i},t\right)} & \;\mathrm{if}\;x_{i}\in \Omega_{obs,trn}\\ 0 & \;\mathrm{otherwise}\;, \end{array}\right.$$

$F_{i}^{3}\left(x_{i},x_{i}^{-},t\right)$ is UAV *i*’s collision avoidance cost,
(20)$$F_{i}^{3}\left(x_{i},x_{i}^{-},t\right)=\frac{1}{N}\sum_{j\neq i}^{N}1_{∥E^{\frac{1}{2}}\left(x_{i}\left(t\right)-x_{i}^{-}\left(t\right)\right)∥\leq e_{0}},$$
$$1_{A}\left(\xi \right)=\left\{\begin{array}{cc} 1 & \;\mathrm{if}\;\xi \in A\\ 0 & \;\mathrm{otherwise}\;, \end{array}\right.$$
where 1 means the indicator function, $e_{0}$ is the safe distance between UAVs, E=diag(1,1,1/c) transforms Euclidean distance into ellipsoidal distance. The interaction cost $F_{i}$ denotes the sum of formation maintenance and trajectory collision loss about UAV *i*.

In summary, the optimal control problem faced by UAV *i* is given by
(21)$$\begin{array}{cc} \; & \underset{u_{i}\in U_{i}}{\inf}J_{i}\left(x_{i},x_{i}^{-},u_{i}\right)\\ \;s.t.\;\; & dx_{i}=h_{i}\left(x_{i},x_{i}^{-},u_{i}\right)dt+\sigma d\omega_{i}\left(t\right). \end{array}$$

To this end, the cooperative control of a large number of UAVs in complex environments is formulated as a noncooperative differential game. Therefore, this is an *N*-player noncooperative game whose well-known solution is the Nash equilibrium (NE), i.e., the control decisions under which no UAV can unilaterally decrease its cost [22].

### 3.2. Mean-Field Game Control Formulation

As the UAV-swarm number *N* increases, the complexity of solving the differential game in (Equation 21) will increase significantly. Mean-field games (MFGs) are a class of problems that encode large populations of interacting agents into systems of coupled partial differential equations, which overcomes the difficulties of large-scale and information sharing. For the current UAV, the neighborhood interaction of formation and collision avoidance, and the environment interaction of obstacle avoidance will cause the UAV to need a great amount of communication resources and computing resources. Traditional optimization and control methods need to separately deal with the increasing interactions, which leads to the problem of dimensionality explosion. Therefore, we reformulate the large-scale UAVs optimal state control problem in complex scenarios as a mean-field game control (MFGC) problem. Under the framework of MFGC, a generic agent only react to the collective behaviors (mean-field) of all agents instead of the behavior of each agent, which greatly reduces the amount of communication and computation. Now, we can drop the index *i* since agents are indistinguishable in MFG. Let ρ(x,t) denote the probability of state x at time *t*; then, the cost functional (Equation 14) is now transformed into
(22)$$J\left(x,\rho ,u\right)=\int_{0}^{T}\left\{\int_{\Omega }L\left(x,u\right)\rho \left(x,t\right)dx+F\left(x,\rho (x,t)\right)\right\}dt+G\left(\rho (\cdot ,T)\right).$$

Meanwhile, the state dynamics in (Equation 11) are transformed into
(23)dx=h(x(t),ρ(x,t),u(x,t))dt+σdω(t).

According to Ito’s lemma [23], (Equation 23) can be represented in terms of the mean field ρ(x,t) and then will be equivalent to the Fokker–Planck equation given by
(24)$$\partial_{t}\rho -\frac{\sigma^{2}}{2}\Delta \rho +\nabla \cdot \left(\rho h\right)=0.$$
with cost functional (Equation 22) and the Fokker–Planck equation in (Equation 24), the high-dimensional MFG which describes the complex-scenario cooperative control of a large number of UAVs is summarized as
(25)$$\begin{array}{cc} \; & \underset{\rho ,u}{\inf}J\left(x(t),\rho (x,t),u(x,t)\right)\\ \;s.t.\; & \partial_{t}\rho -\frac{\sigma^{2}}{2}\Delta \rho +\nabla \cdot \left(\rho h\right)=0\\ \; & \;\;\;\rho \left(x,0\right)=\rho_{0}\left(x\right), \end{array}$$
where $\rho_{0}\left(x\right)$ is the initial probability distribution of the UAV swarm’s states. To this end, the state control for a large number of UAVs is formulated as a high-dimensional MFG. The agents forecast a distribution of the population ${\left\{\rho \left(\cdot ,t\right)\right\}}_{t=0}^{T}$ and aim at minimizing their cost. Therefore, at a Nash equilibrium, we have that, for every $x\in R^{n}$,
(26)$$J\left(x,\rho ,\hat{u}\right)\leq J\left(x,\rho ,u\right),\;\forall u\in U,$$
where $\hat{u}$ is the equilibrium strategy of an agent at state x. Here, we assume that agents are small, and their unilateral actions do not alter the density ρ [14]. With finite UAVs, it yields an MF approximation that achieves the ϵ-NE [10,22].

### 3.3. GAN-Based Algorithm for Complex-Scenario MFGs

In this section, we propose a generative-adversarial-network (GAN)-based approach to solving the high-dimensional MFG in (Equation 25). Inspired by Wasserstein GANs [24], APAC-Net [15,25], we use the variational primal–dual structure of MFG and phrase (Equation 25) as a convex–concave saddle-point problem. Then, we propose a GAN-based algorithm to solve the complex-scenario MFG in (Equation 25).

Now, we show the underlying primal–dual structure of the complex-scenario MFG and derive the convex–concave saddle-point problem equivalent to (Equation 25). Denote ϕ as the Lagrange multiplier, we can put the differential constraint in (Equation 24) (Fokker-Planck equation) into the cost function in (Equation 22) to obtain the following extended cost function
(27)$$\underset{\phi }{\sup}\underset{\rho ,u}{\inf}\int_{0}^{T}\left\{\int_{\Omega }\rho \left(x,t\right)L\left(x,u(x,t)\right)dx+F\left(\rho (\cdot ,t)\right)\right\}dt+G\left(\rho (\cdot ,T)\right)-\int_{0}^{T}\int_{\Omega }\phi \left(x,t\right)\left(\partial_{t}\rho -\frac{\sigma^{2}}{2}\Delta \rho +\nabla \cdot \left(\rho \left(x,t\right)h\left(x,t\right)\right)\right)dxdt.$$

Minimizing concerning u to get the Hamiltonian via $H\left(x,p\right)=\inf_{u}\left\{ph+L\left(x,u\right)\right\}$ and integrating by parts, we can rewrite (Equation 27) as
(28)$$\underset{\rho }{\inf}\underset{\phi }{\sup}\int_{0}^{T}\left\{\int_{\Omega }\left(\partial_{t}\phi +\frac{\sigma^{2}}{2}\Delta \phi +H\left(x,\nabla \phi \right)\right)\rho \left(x,t\right)dx+F\left(\rho (\cdot ,t)\right)\right\}dt+\int_{\Omega }\phi \left(x,0\right)\rho_{0}\left(x\right)dx-\int_{\Omega }\phi \left(x,T\right)\rho \left(x,T\right)dx+G\left(\rho \left(\cdot ,T\right)\right).$$

The Formular (Equation 28) is the cornerstone of our algorithm.

We solve (Equation 28) by training a GAN-like neural network. Although the idea is inspired by Wasserstein GANs [24] and APAC-Net [15,25], the loss function of our algorithm considers the formation similarity $F^{1}$ and dense obstacle avoidance $F^{2}$, which provides greater flexibility for large-scale agents maneuvering; details are shown in Algorithm 1. We show the structure and training process of our GAN-based neural network in Figure 1.
**Algorithm 1** GAN-based algorithm for complex-scenario MFGs**Require:**σ diffusion parameter, *H* Hamiltonian, *g* terminal cost, *f* interaction term.**Require:** Initialize neural networks $N_{\omega }$ and $N_{\theta }$, batch size *B*.**Require:** Set $\phi_{\omega }$ and $G_{\theta }$ as in (Equation 29).**while** not converged **do**         **train** $\phi_{\omega }$:         Sample batch ${\left\{\left(z_{b},t_{b}\right)\right\}}_{b=1}^{B}$ where $z_{b}∼\rho_{0}$ and $t_{b}∼\mathrm{Unif}\left(0,T\right)$         Obtaining generated data ${\left\{\left(x_{b},t_{b}\right)\right\}}_{b=1}^{B},x_{b}\leftarrow G_{\theta }\left(z_{b},t_{b}\right)$.         Update discriminator parameters ω to minimize $\ell_{\mathrm{total}\;}=\ell_{0}+\ell_{t}+\ell_{\mathrm{HJB}}$                  $\omega \leftarrow \omega -\eta_{1}\nabla \ell_{\mathrm{total}\;}\left(\omega \right)$         **train** $G_{\theta }:$         Sample batch ${\left\{\left(z_{b},t_{b}\right)\right\}}_{b=1}^{B}$ where $z_{b}∼\rho_{0}$ and $t_{b}∼\mathrm{Unif}\left(0,T\right)$         Update generator parameters θ to minimize $\zeta_{t}$                  $\theta \leftarrow \theta -\eta_{2}\nabla \zeta_{t}\left(\theta \right)$**end while**

First, we initialize the neural networks $N_{\omega }\left(x,t\right)$ and $N_{\theta }\left(z,t\right)$, then set
(29)$$\phi_{\omega }\left(x,t\right)=\left(1-t\right)N_{\omega }\left(x,t\right)+tg\left(x\right),G_{\theta }\left(z,t\right)=\left(1-t\right)z+tN_{\theta }\left(z,t\right),$$
where $z∼\rho_{0}$ are samples drawn from the initial distribution. The formulation of $\phi_{\omega }$ and $G_{\theta }$ in (Equation 29) automatically satisfy the terminal and initial condition, respectively. Moreover, our algorithm encodes the underlying structure of complex-scenario MFGs via (Equation 28) and (Equation 29), which absolves the neural networks learning for the entire solution of the MFG from scratch.

Our approach for training neural networks includes parallel alternately training $G_{\theta }$ (the state density distribution) and $\phi_{\omega }$ (the value function). In order to obtain the equilibrium of the complex-scenario MFG, we train $\phi_{\omega }$ by first sampling a batch ${\left\{z_{b}\right\}}_{b=1}^{B}$ from $\rho_{0}$, and ${\left\{t_{b}\right\}}_{b=1}^{B}$ uniformly from [0,T]. Then, we compute the push-forward states $x_{b}=G_{\theta }\left(z_{b},t_{b}\right)$ for b=1,⋯,B. The main loss item for training the discriminator $\phi_{\omega }$ is
(30)$${\mathrm{loss}}_{\phi }=\underset{\ell_{0}}{\underset{︸}{\frac{1}{B}\sum_{b=1}^{B}\phi_{\omega }\left(x_{b},0\right)}}+\underset{\ell_{t}}{\underset{︸}{\frac{1}{B}\sum_{b=1}^{B}\partial_{t}\phi_{\omega }\left(x_{b},t_{b}\right)+\frac{\sigma^{2}}{2}\Delta \phi_{\omega }\left(x_{b},t_{b}\right)+H\left(\nabla_{x}\phi_{\omega }\left(x_{b},t_{b}\right),x_{b}\right)}},$$
where we can optionally add a regularization term
(31)$$\ell_{\mathrm{HJB}}=\lambda \frac{1}{B}\sum_{b=1}^{B}∥\partial_{t}\phi_{\omega }\left(x_{b},t_{b}\right)+\frac{\sigma^{2}}{2}\Delta \phi_{\omega }\left(x_{b},t_{b}\right)+H\left(\nabla_{x}\phi_{\omega }\left(x_{b},t_{b}\right),x_{b}\right)+f\left(x_{b},t_{b}\right)∥$$
to penalize deviations from the HJB equations. This extra regularization term has also been found effective in, e.g., Wasserstein GANs. Finally, we backpropagate the total loss to update the weights of the discriminator $\phi_{\omega }$. To train the generator, we again sample ${\left\{z_{b}\right\}}_{b=1}^{B}$ and ${\left\{t_{b}\right\}}_{b=1}^{B}$ as before. The loss of the generator is given by
(32)$${\mathrm{loss}}_{G}=\underset{\zeta_{t}}{\underset{︸}{\frac{1}{B}\sum_{b=1}^{B}\partial_{t}\phi_{\omega }\left(G_{\theta }\left(z_{b},t_{b}\right),t_{b}\right)+\frac{\sigma^{2}}{2}\Delta \phi_{\omega }\left(G_{\theta }\left(z_{b},t_{b}\right),t_{b}\right)+H\left(\nabla_{x}\phi_{\omega }\left(G_{\theta }\left(z_{b},t_{b}\right),t_{b}\right),x_{b}\right)+f\left(G_{\theta }\left(z_{b},t_{b}\right),t_{b}\right)}}.$$

At last, we backpropagate this loss to update the weights of $G_{\theta }$. In conclusion, in each time slot t∈[0,T], the neural networks will be trained. The generator $G_{\theta }$ will generate the state distribution at time *t* and the discriminator $\phi_{\omega }$ will get the result of the value function at time *t*.

## 4. Simulation Results

In this section, to verify the effectiveness and robustness of our method, we conduct a series of simulation experiments of distributed aerial large-scale UAVs. To demonstrate the competitiveness of our method, we compare it against baseline methods regarding their performances at last.

### 4.1. Simulation Parameters

For the model hyperparameters, we set $c_{1}=0.5$ (the weight of the control effort in (Equation 15)), $c_{2}=5$ (the weight of the terminal cost in (Equation 16)), and $l_{1}=5$, $l_{2}=10$, $l_{3}=5$ (the weights of the interaction cost in (Equation 17)). For the neural networks, there are three linear hidden layers with 100 hidden units per layer in both networks. Residual neural networks (ResNet) are used for both networks, with a skip connection weight of 0.5. Tanh activation function is used in $\phi_{\omega }$ while ReLU activation function is used in $G_{\theta }$. We choose ADAM as the optimizer with β=(0.5,0.9), $\eta_{1}=4\times 10^{-4}$ (the learning rate for $\phi_{\omega }$), $\eta_{2}=1\times 10^{-4}$ (the learning rate for $G_{\theta }$), weight decay of $10^{-4}$ for both networks, batch size 25, and λ=1 (the weight of the HJB regularization term in (Equation 31)).

### 4.2. Formation Performance of Large-Scale UAVs

To validate the feasibility and robustness of our method for large-scale UAVs formation, we simulated twenty-five drones flying in desired formations from the bottom left side to the top right side in the 3-D scene, as shown in Figure 2. We experimented with two forms of formations—spiral formation and wave formation, which could confirm the generality of our approach. In Figure 2a, we set the initial points of the drones as $\left(2cos\left(\frac{2\pi i}{25}\right),2sin\left(\frac{2\pi i}{25}\right)+18,\frac{5i}{24}-10\right)$, i=0,1,...,24; the target points of the drones are $\left(2cos\left(\frac{2\pi i}{25}\right),2sin\left(\frac{2\pi i}{25}\right)-18,\frac{5i}{24}+5\right)$, i=0,1,...,24. In Figure 2b, the initial points of the drones are $\left(\frac{20i}{25}-10,sin\left(\frac{20i}{25}-10\right)+18,\frac{5i}{24}-10)\right)$, i=0,1,...,24; the target points of the drones are $\left(\frac{20i}{25}-10,sin\left(\frac{20i}{25}-10\right)-18,\frac{5i}{24}+5)\right)$, i=0,1,...,24.

The test results are shown in Figure 3, which demonstrates that our method can maintain the formation of large-scale swarms in 3-D environments. To certify the generality of our approach in large-scale formations, we design two forms of formations consisting of twenty-five quadrotors—spiral and wave formations. As depicted in Figure 3a,b, the xoy -plane projections illustrate the shape contour of the formation maintenance, which means that the swarm maintains the desired formation well during the flight.

The convergence of our proposed MFG control algorithm and the effectiveness and stability of formation control under different forms of formations are shown in Figure 4. In Figure 4a, we plot the HJB residual errors, i.e., $\ell_{\mathrm{HJB}}$ in Algorithm 1, which shows the convergence of MFGC. Without an efficient strategy control, the HJB residuals under different formations are relatively high. HJB residuals drop fast after we apply a series of controls. After around $5\times 10^{4}$ iterations, the error curves tend to be stable when we obtain an optimal control for UAVs. In Figure 4b, we plot the formation loss. Without a proper design of the formation cost term, there is no effective formation control, and the formation loss under different formations is relatively high. Formation loss drops fast after we apply our formation cost term. After around $1\times 10^{5}$ iterations, the error curves tend to be stable when we obtain an optimal formation control for UAVs.

### 4.3. Effect of Volatility Term σ on Formation

We investigate the effect of the volatility term σ on the behavior of the MFGC solutions—the executed trajectories for the formation flight of large-scale UAVs. In Figure 5, we show the solutions for MFGCs using σ=0, $\sqrt{2}\times 10^{-1}$ and $\sqrt{10}\times 10^{-1}$. As σ increases, the density of UAVs widens along the paths, and the desired formation maintenance of UAVs decreases due to the added diffusion term in the MFG control equations in (Equation 25). From Table 1, it can be seen that larger volatility σ corresponds to greater formation cost $F^{1}$, but the collision avoidance cost $F^{3}$ is almost invariant. These results are consistent with those in Figure 5.

### 4.4. Performance of Dense Obstacle Avoidance for Large-Scale UAVs Formation Flying

To verify the effectiveness of our method with large-scale UAVs formation flying through an obstacle-rich area, we design a spiral formation consisting of twenty-five quadrotors. A packed area of nine cuboid obstacles is set up in the simulation. As depicted in Figure 6, the swarm successfully avoids the obstacles and the desired formation is well preserved during the flight. Under our MFG control, the formation similarity error is steadily close to 0 as shown in Figure 7a; the executed trajectories are zero-collision, which can be seen from Figure 7b,c.

### 4.5. Comparison with Baselines

In the last part of the simulation results, we verify the superiority of our method by comparing it to some typical baseline methods in Table 2. From Table 2, it can be seen that our approach simplifies large-scale communication and handles the most complex application scenarios. For more details of these baseline methods, please refer to the related documents [4,7,9,13], etc.

## 5. Conclusions

In this paper, we propose a large-scale agent trajectory optimization method capable of navigating large-scale swarms in formations while avoiding dense obstacles. First, two types of differentiable mean-field terms are developed, where the overall similarity distance between large-scale 3-D formations and the potential energy value of dense 3-D obstacles are quantified respectively. Second, we embed these two terms into a designed mean-field game control framework that simultaneously satisfies the dynamic constraints and minimizes energy cost, formation similarity error, and collision penalty. This framework solves the communication and computing difficulties caused by complex scenarios of large-scale multiagent systems and formulates the trade-offs between formation and obstacle avoidance in large-scale contexts. Finally, the solid performance of our method in simulations of the formation and dense obstacle avoidance for large-scale UAVs validates its practicality and efficiency. The comparison with baseline methods shows the advanced nature of our method. In the future, we will consider the application and adaption of this mean-field game control framework in more complex scenarios of multiagent systems, such as complex communication conditions (time-varying, partial rejection) and dense mixed obstacles (static, dynamic).

## Figures and Tables

**Figure 1 sensors-22-05437-f001:**
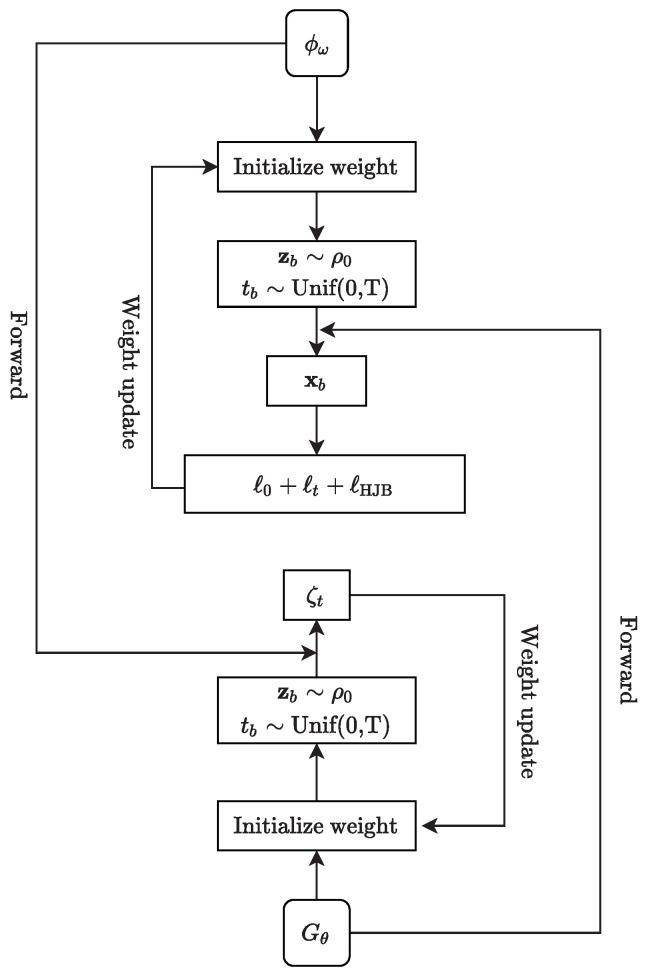
Visualization of the structure and training process of our GAN-based neural network. Its training process is divided into two coupled alternating training parts—generator and discriminator.

**Figure 2 sensors-22-05437-f002:**
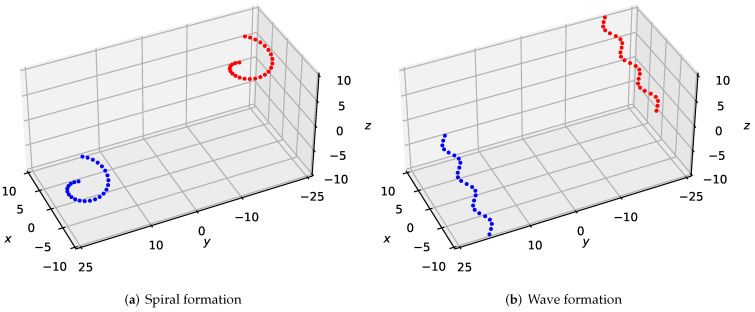
A large-scale desired formation consisting of twenty-five quadrotors traverses a 3-D environment from the bottom left side to the top right side.

**Figure 3 sensors-22-05437-f003:**
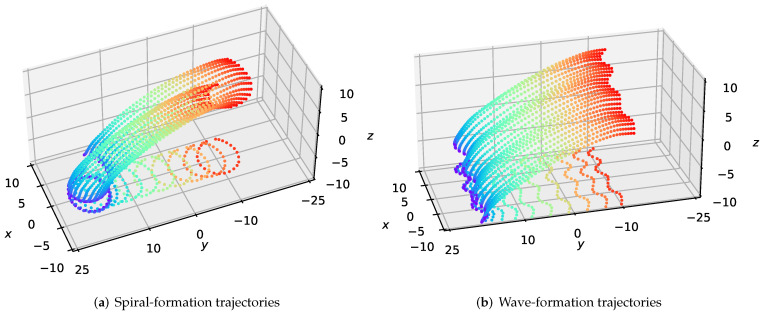
The visualization of the executed trajectories for the formation flight of large-scale UAVs. The xoy-plane projections represent the outline of the shape.

**Figure 4 sensors-22-05437-f004:**
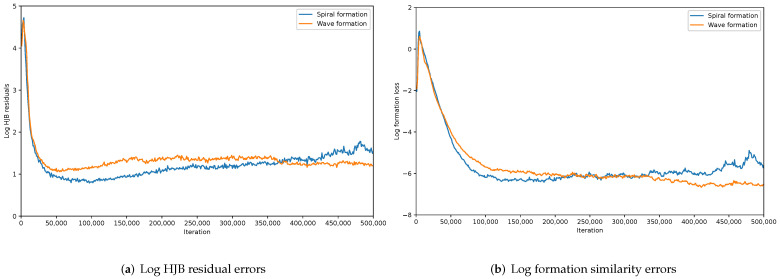
Illustration of MFG convergence and formation stability.

**Figure 5 sensors-22-05437-f005:**
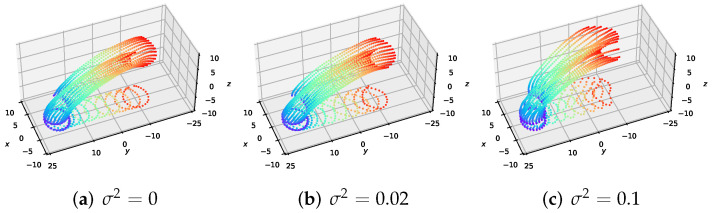
Comparison of the executed trajectories for the formation flight of large-scale UAVs about volatility parameter σ=0, $\sqrt{2}\times 10^{-1}$ and $\sqrt{10}\times 10^{-1}$.

**Figure 6 sensors-22-05437-f006:**
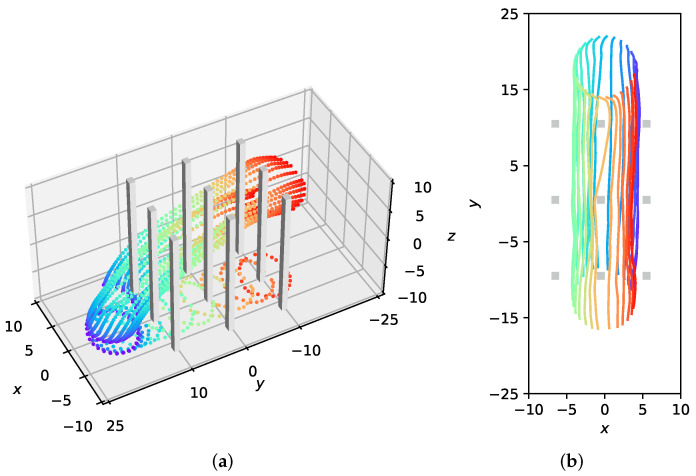
The visualization of the executed trajectories for large-scale UAVs formation flying through an obstacle-rich area. (**a**) Full view of the trajectories. (**b**) xoy-plane projection trajectories.

**Figure 7 sensors-22-05437-f007:**
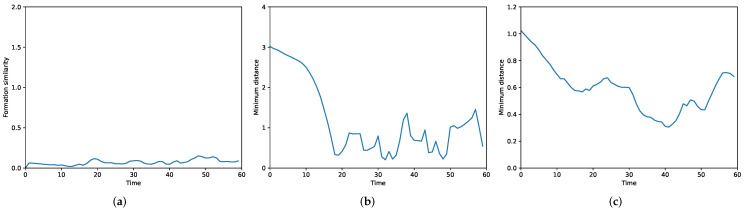
Illustration of the performance of interaction terms for MFG control ($F^{1}$, $F^{2}$, $F^{3}$). (**a**) Formation similarity error. (**b**) Minimum distance between UAVs and obstacles. (**c**) Minimum distance between UAVs.

**Table 1 sensors-22-05437-t001:** Comparison with interaction costs.

Volatility Parameter σ	Formation Cost $F^{1}$	Collision Avoidance Cost $F^{3}$
0	$1.41\times 10^{-3}$	$6.35\times 10^{-2}$
$\sqrt{2}\times 10^{-1}$	$6.34\times 10^{-2}$	$6.34\times 10^{-2}$
$\sqrt{10}\times 10^{-1}$	$4.30\times 10^{-1}$	$6.14\times 10^{-2}$

**Table 2 sensors-22-05437-t002:** Comparison with baseline methods.

Method	Scene	Scale of UAVs	Scene Complexity	Communication
[4]	Formation flight and obstacle avoidance	Small	0.67 ^1^	O(N) ^2^
[9]	Cluster flight	Large	0.5	O(1)
[13]	Cluster flight and obstacle avoidance	Large	0.67	O(1)
[7]	Formation flight and dense obstacle avoidance	Small	0.83	O(N)
Ours	Formation flight and dense obstacle avoidance	Large	1	O(1)

^1^ The measurement method of scene complexity is as follows: here it is a scoring system, [cluster flight, formation flight] = [1′, 2′]; [obstacle avoidance, dense obstacle avoidance] = [1′, 2′]; [small, large] = [1′, 2′]. We accumulate the scores for each literature experiment scene according to each item and finally normalize them. ^2^
O() is infinitesimal of the same order.

## Data Availability

Not applicable.

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
