# Peer review of "A Mean-Field Game Control for Large-Scale Swarm Formation Flight in Dense Environments"

_sensors, 2022, doi:10.3390/s22145437_

Round 1

Reviewer 1 Report

This manuscript introduces a mean-field game control method for large-scale swarm formation flight. A general control framework and corresponding mean-field terms are contributed to solve the large-scale swarm formation flight trajectoryFurthermore, GAN-based algorithm is proposed to address complex-scenario MFGs problem. This method is potential in large-scale UAV control mission. However, some problems still exist in the manuscript. A revision is required before publication.

Minor:

(1)The manuscript needs to be carefully checked and proofread. There are too many long sentences in the manuscript. These sentences are hard to read and understand, and easy to make the reader confused;

(2)Some figures are unlabeled, such as Fig. 1, 3 and so on;

(3)P3, “A series of simulations with…” is a validation, but not a contribution;

(4)Eqs.(8), (26), (27) and (31) are not in the correct format;

Major:

(1)Mean-Field Game Control Formulation is one of the main innovations of this paperHowever, the relevant description is very briefPlease add detail descriptions about the practical scenario of the mean-field game;

(2)Isection 3.3more details of the GAN and samples should be added. A figure is required to illustrate the framework of the GAN-based algorithm, inputs and outputs;

(3)P13, Table 2, what is the definition of “scale” and “Scene complexity”?

(4)Simulations, more analysis about the control characteristics and swarm behaviors can be carried out, or this part seems shallow.

Round 2

Reviewer 1 Report

accept as is